# Repat33 Acts as a Downstream Component of Eicosanoid Signaling Pathway Mediating Immune Responses of *Spodoptera exigua*, a Lepidopteran Insect

**DOI:** 10.3390/insects12050449

**Published:** 2021-05-14

**Authors:** Md Tafim Hossain Hrithik, Mohammad Vatanparast, Shabbir Ahmed, Yonggyun Kim

**Affiliations:** Department of Plant Medicals, College of Life Sciences, Andong National University, Andong 36729, Korea; hrithik11605@gmail.com (M.T.H.H.); mvatanparast@korea.kr (M.V.); shabbirahmed.du@gmail.com (S.A.)

**Keywords:** immunity, repat, eicosanoid, PLA_2_, nodulation, AMP, *Spodoptera exigua*

## Abstract

**Simple Summary:**

Eicosanoids are oxygenated polyunsaturated fatty acids containing 20 carbons and subdivided into prostaglandin, leukotriene, and epoxyeicosatrienoic acid. They mediate immune responses in insects as well as mammals. Excessive eicosanoids cause severe inflammatory diseases in humans. However, a lack of eicosanoids induces immunosuppressive conditions in insects, which are highly susceptible to various entomopathogens. Despite the physiological significance of eicosanoids, their molecular action was not clearly understood in insects. This study was focused on Repat (Response to Pathogen) gene family, which might be regulated by eicosanoids. Among 44 members of Repat family genes, Repat33 was highly inducible to Gram-negative bacteria. Its expression was dependent on eicosanoids. Loss of function of Repat33 by RNA interference (RNAi) caused significant immunosuppression of a lepidopteran insect, *Spodoptera exigua*. Larvae of S. exigua treated by RNAi did not exhibit efficient cellular immune responses. They also failed to express antimicrobial peptide genes at high inducible levels in response to bacterial challenges. These results suggest that Repat33 is a downstream component of eicosanoid immune mediation.

**Abstract:**

Repat (=response to pathogen) is proposed for an immune-associated gene family from *Spodoptera exigua*, a lepidopteran insect. In this gene family, 46 members (*Repat1*–*Repat46*) have been identified. They show marked variations in their inducible expression patterns in response to infections by different microbial pathogens. However, their physiological functions in specific immune responses and their interactions with other immune signaling pathways remain unclear. *Repat33* is a gene highly inducible by bacterial infections. The objective of this study was to analyze the physiological functions of *Repat33* in mediating cellular and humoral immune responses. Results showed that *Repat33* was expressed in all developmental stages and induced in immune-associated tissues such as hemocytes and the fat body. RNA interference (RNAi) of *Repat33* expression inhibited the hemocyte-spreading behavior which impaired nodule formation of hemocytes against bacterial infections. Such RNAi treatment also down-regulated expression levels of some antimicrobial genes. Interestingly, *Repat33* expression was controlled by eicosanoids. Inhibition of eicosanoid biosynthesis by RNAi against a phospholipase A_2_ (PLA_2_) gene suppressed *Repat33* expression while an addition of arachidonic acid (a catalytic product of PLA_2_) to RNAi treatment recovered such suppression of *Repat33* expression. These results suggest that Repat33 is a downstream component of eicosanoids in mediating immune responses of *S. exigua*.

## 1. Introduction

Insects can defend against microbial pathogens through their immune responses [1]. Insect innate immunity involves programmed immune responses that are activated upon a pathogen attack [2]. To be specific, pathogen recognition receptors can recognize specific pathogen-associated molecular patterns upon immune challenge [3]. The recognition signal is then relayed to nearly all immune tissues such as hemocytes and fat body via various immune mediators [4]. These immune effectors can express cellular immune responses such as phagocytosis, nodule formation, encapsulation [5], and humoral immune responses such as melanization and antimicrobial peptide (AMP) gene expression [6].

Repat (response to pathogen) is an immune-associated gene family encoded in *Spodoptera exigua*, a lepidopteran insect [7]. Although Repat genes were initially discovered in the larval midgut after infection by *Bacillus thuringiensis* (Bt), their expression levels were also up-regulated by baculovirus infection [8]. Interestingly, some Repat genes are highly expressed in a Bt-resistant strain [8]. Repat proteins are relatively small (≈15 kDa). They can interact with other Repat proteins [9]. For example, Repat1 is localized in the cytosol but translocated into the nucleus in the presence of Repat8 [10]. However, the functional associations of Repat proteins with immune responses are little understood.

Eicosanoids are a group of oxygenated C20 polyunsaturated fatty acids [11]. Eicosanoid biosynthesis is triggered by the hydrolysis of fatty acids for phospholipids using phospholipase A_2_ (PLA_2_) [12]. In *S. exigua*, three different PLA_2_s have been identified. They play crucial roles in mediating immune functions [13,14,15]. Especially, secretory PLA_2_ (sPLA_2_) exhibits diverse roles in the digestion of dietary lipid in the midgut and in the production of eicosanoids to mediate immune responses [15,16]. Eicosanoid biosynthesis in insects is similar to that of mammalian systems but exhibits some deviation. Phospholipids in terrestrial insects have a little amount of arachidonic acid (AA) [17]. Thus PLA_2_ catalyzes phospholipids to release linoleic acid which is then elongated and desaturated to form AA in *S. exigua* [18]. AA is then oxygenated by cyclooxygenase-like peroxynectin to produce PGH_2_, a prostaglandin (PG) precursor [13]. Finally, different PG synthases such as PGE_2_, PGD_2_, and PGI_2_ in *S. exigua* [19,20,21] can isomerize PGH_2_. Thromoboxane B_2_ is another type of prostanoid identified in *S. exigua* [22]. AA can also be oxygenated by different epoxygenases into four epoxyeicosatetraenoic acids in *S. exigua* [23]. Although leukotrienes are not identified in insects, they are known to play crucial roles in mediating immune responses in *S. exigua* [24]. All these eicosanoids can mediate immune responses and reproductive processes in *S. exigua* [4].

With the discovery of the PGE_2_ receptor from *Manduca sexta*, another lepidopteran insect, the eicosanoid immune signaling pathway has been investigated [25]. A similar PGE_2_ receptor of *S. exigua* can also use cAMP secondary messenger to activate actin cytoskeleton rearrangement during hemocyte-spreading behavior in cellular immune responses [26]. These findings suggest that other downstream components of the PGE_2_ immune signaling pathway are activated to mediate cellular and humoral immune responses. 

Inducible expression of Repat family genes in response to immune challenge led us to impose a hypothesis that they might be downstream components of the eicosanoid immune signaling pathway. To test this hypothesis, we selected a specific Repat gene, *Repat33*, which was highly inducible by bacterial infection, and assessed its immune-associated function using RNA interference (RNAi) by monitoring any loss of functions. Then we investigated the functional link between eicosanoid signaling and *Repat33* expression via PLA_2_.

## 2. Materials and Methods

### 2.1. Insect Rearing 

*S. exigua* larvae were collected from a Welsh onion field in Andong, Korea, and reared with an artificial diet [27] at 25 ± 1 °C. They underwent five instars (L1–L5). Sugar solution (10%) was used as an adult diet. 

### 2.2. Immune Challenge

*Escherichia coli* Top10 Gram-negative bacterium was obtained from Invitrogen (Carlsbad, CA, USA) and cultured on Luria-Bertani (LB) medium. For the immune challenge, bacteria were killed by heat treatment at 95 °C for 10 min. Not a single colony of heat-treated bacteria grew on LB medium. *Enterococcus mundtii*, a Gram-positive bacterium, was isolated from a nematode [28]. It was cultured with tryptic soy broth and heat-killed for the immune challenge. *Autographa californica* multiple nucleopolyhedrosis virus (AcMNPV), a baculovirus, was cultured using Sf9 cells according to the method described by Jung and Kim [29]. *Metarhizium rileyi*, an entomopathogenic fungus, was isolated from infected *S. exigua* in a welsh onion field [30] and cultured on solid potato dextrose agar medium (5% potato extract, 0.5% dextrose, 1.7% agar) at 25 °C for 7 days. Conidia suspension was prepared by scraping the fungal culture into 0.1% Triton X−100 solution and counted using a Neubauer hemocytometer (Marienfeld-Superior, Lauda-Königshofen, Germany) under 40× magnification. For the immune challenge, L5 larvae were injected in 1 µL of bacteria (10^4^ cells/larva), 1000 conidia of *M. rileyi*, or AcMNPV in a budded form (10^4^ plaque-forming unit (pfu)/larva).

### 2.3. Chemicals

Arachidonic acid (AA: 5,8,11,14-eicosatetraenoic acid) and dexamethasone (DEX:(11β,16α)−9-fluoro−11,17,21-trihydroxy−16-methylpregna−1,4-diene−3,20-dione) were purchased from Sigma-Aldrich Korea (Seoul, Korea). Both were dissolved in dimethyl sulfoxide (DMSO). Diethyl pyrocarbonate (DEPC) water was prepared by mixing 1 mL of DEPC with 1 L of deionized distilled water followed by 12 h incubation at 37 °C. DEPC-Treated water was then autoclaved. Anticoagulant buffer was prepared with 186 mM NaCl, 17 mM Na_2_EDTA, and 41 mM citric acid. Its pH was then adjusted to 4.5 with HCl. Phosphate-buffered saline (100 mM phosphate containing 0.7% NaCl, pH 7.4) was prepared. 

### 2.4. Larval Tissue Preparation

Three-day-old L5 larvae were used to collect different tissues including hemocytes, fat body, midgut, and epidermis. Hemolymph was collected by cutting prolegs and centrifuged at 800× *g* for 5 min to obtain a hemocyte pellet. After hemolymph collection, larvae were used to isolate other tissues such as fat body, midgut, and epidermis. Each tissue sample was prepared from five L5 larvae and replicated three times.

### 2.5. RNA Extraction and cDNA Preparation

From different developmental stages and different tissues, total RNAs were extracted using Trizol reagent (Invitrogen). RNA was resuspended in DEPC water and heated at 50 °C for 15 min. RNA concentration was measured with a spectrophotometer (Nanodrop, Thermo Fisher Scientific, Wilmington, DE, USA). For cDNA preparation, 2 μg of RNA was mixed with oligo-dT and reverse transcriptase (Intron Biotechnology, Seoul, Korea).

### 2.6. Bioinformatics Analysis 

*S. exigua* Repat gene sequences were obtained from the transcriptomic databases (PRJNA634227). Prediction of protein domain structure was performed using Pfam (http://pfam.xfam.org, accessed on 14 May 2020) and Prosite (https://prosite.expasy.org/, accessed on 14 May 2020).

### 2.7. RT-PCR and RT-qPCR

RT-PCR was performed using DNA Taq polymerase (GeneAll, Seoul, Korea) with gene-specific primers (Appendix A) and a sequential temperature cycle of 95 °C for 30 s, 52 °C for 30 s, and 72 °C for 30 s. After 35 cycles, the PCR product was assessed by 1% agarose gel electrophoresis. RT-qPCR was performed using a CFXC connect real-time PCR Detection System (BioRad, Hercules, CA, USA) with a SYBER Green Real-Time PCR Master Mix (Toyobo, Osaka, Japan) according to the guideline provided by Bustin et al. [31]. A ribosomal protein, *RL32*, was used as a stably expressed reference gene [13].

### 2.8. RNA Interference (RNAi)

T7 promoter sequence was attached to gene-specific primers at the 5′ end. Using these primers, a partial gene (297 bp) of *Repat33* was amplified with the method described above. Using this PCR product, double-stranded RNA (dsRNA) was prepared with a Megascript RNAi Kit (Ambion, Austin, TX, USA). dsRNA was combined with a transfection reagent (Metafectene Pro, Biontex, Planegg, Germany) at a 1:1 ratio. A microsyringe (Hamilton, Reno, NV, USA) equipped with a 26-gauge needle was used to inject the dsRNA sample to L5 larvae at a dose of 1 µg dsRNA per larva. RNA efficiency was estimated by RT-qPCR at 48 h post-injection (PI). Each treatment was replicated three times with independent RNA preparation. For RNAi of *Se-sPLA_2_*, we followed the method described previously [15].

### 2.9. Hemocyte-Spreading Behavior Analysis

L5 larvae were treated with RNAi. After 24 h, they were injected with *E. coli* (10^4^ cells/larva). After 8 h of incubation at 25 °C, approximately 100 µL of hemolymph was obtained and mixed with 800 µL of the anticoagulant buffer. After 5 min of incubation on ice, the cell suspension was centrifuged at 800× g for 5 min. Then 700 µL of the supernatant was replaced with 250 µL of TC100 insect culture medium (Welgene, Gyeongsan, Korea) to prepare a hemocyte suspension. For quantification of hemocyte-spreading behavior, 50 µL of the hemocyte suspension was added to each well of a 96-well microplate and incubated at 25 °C for 40 min under darkness. Spread hemocytes were determined from randomly chosen 100 hemocytes by cytoplasmic extension beyond cell boundary. For F-actin analysis, 10 µL of the hemocyte suspension was used to observe cells after incubation in a wet chamber under darkness. After removing the supernatant, attached cells were fixed with 3.7% paraformaldehyde at room temperature (RT) for 10 min. After washing three times, cells were permeabilized with PBS containing Triton X−100 for 2 min. These permeabilized cells were then incubated with 5% skimmed milk to block the background for 10 min at RT. Cells were then incubated with fluorescein isothiocyanate (FITC)-tagged phalloidin (Alexa Fluor 488 phalloidin, 1 µg/mL, Thermo Scientific, Rockford, IL, USA) in PBS at RT for 1 h. Before mounting slides, cells were stained with 4′,6-diamidino−2-phenylindole (DAPI, 1 µg/mL, Thermo Scientific) in PBS. Hemocyte-spreading was observed under a fluorescence microscope (DM2500, Leika, Wetzlab, Germany). Hemocyte-spreading was determined by the extension of F-actin out of the original cell boundary. Hemocyte-spreading behavior was quantified by randomly assessing 100 cells. Each treatment was replicated three times.

### 2.10. Nodulation Assay

Two-day-old L5 larvae were used for hemocyte nodule formation bioassays. Heat-killed *E. coli* was injected at a dose of 1.63 × 10^4^ cells per larva through proleg and incubated at RT for 8 h. For RNAi, bacteria were injected at 48 h PI of dsRNA. After dissection of test larvae, melanized nodules were counted under a stereoscopic microscope (Stemi SV11, Zeiss, Jena, Germany) at 50× magnification. Each treatment was replicated three times.

### 2.11. Expression Analysis of Antimicrobial Peptide (AMP) Genes

Expression levels of AMP genes were analyzed using L5 larvae after immune challenge with heat-killed *E. coli* as described above. At 8 h after immune challenge, different tissues were prepared and used to extract total RNAs. RT-qPCR was performed as described above against 10 AMP genes with gene-specific primers (Appendix A). Each treatment was replicated three times.

### 2.12. Data Analysis

All data were analyzed using PROC GLM of the SAS program [32]. Means were compared by LSD test at Type I error = 0.05.

## 3. Results

### 3.1. Repat Family Is Subdivided into Three Groups

Repat family was initially reported in the transcriptome of *S. exigua* (GenBank accession number: PRJNA634227), in which 46 members were deposited in NCBI GenBank. These Repat genes were phylogenetically classified into three groups (Figure 1a). Group I was the largest subfamily. It contained 24 members. This group was further subdivided into four clusters: Ia, Ib, Ic, and Id. Group II contained 17 members. Group III contained only five Repat genes. *Repat33* was classified into Group Ia along with *Repat31* and *Repat32*. 

Repat genes showed various inducible expression patterns after challenges with different pathogens (Figure 1b). Some Repat genes in Groups II and III were inducible after a viral infection. In contrast, some Repat genes in Group I was inducible after a bacterial infection as well as after a viral infection. However, *Repat33* was only highly inducible after infection of Gram-negative bacteria and used for subsequent functional assay in response to the bacterial immune challenge.

### 3.2. Expression Profile of Repat33

Based on the predicted amino acid sequence of *Repat33*, its product is a small (110 amino acid residues) secretory protein with a signal peptide in its amino-terminal region. It also possessed a transcription factor domain (Figure 2a). It was expressed in all developmental stages, from the egg stage to the adult stage (Figure 2b). Especially, it was highly expressed in the L5 larval stage. Four different tissues (midgut, epidermis, fat body, and hemocytes) expressed *Repat33* at the L5 stage. In response to bacterial immune challenge, all tested tissues showed significantly (*p* < 0.05) increased expression of *Repat33*.

### 3.3. RNAi of Repat33 and Subsequent Influence on Immune Responses

RNAi was performed by injecting 1 μg of dsRNA specific to *Repat33* to larval hemocoel of *S. exigua* (Figure 3a). Expression levels of *Repat33* were significantly (*p* < 0.05) reduced by dsRNA injection. At 24 h after dsRNA injection, the expression level of *Repat33* was reduced by more than 90%. After that, its expression level began to increase, fully recovering to control level at 72 h after dsRNA injection.

Under this RNAi condition, hemocytes significantly lost their spreading behavior (Figure 3b). When the actin cytoskeleton was observed after staining F-actin using FITC-labeled phalloidin, control hemocytes exhibited extensive cytoplasmic extension while hemocytes collected from RNAi-treated larvae did not. The number of nodules formed in response to the bacterial challenge was significantly (*p* < 0.05) decreased after treatment with *Repat33* dsRNA compared to that after treatment with dsRNA control (Figure 3c).

Humoral immune response was assessed by measuring expression levels of 10 AMP genes in four tissues: fat body (‘FB’), hemocytes (‘HC’), midgut (MG’), and epidermis (‘EP’) (Figure 4). In naïve larvae, the AMP expression profile varied among tissues. In the fat body, *lysozyme* was highly expressed while attacin−1 in hemocytes, *defensin* in the midgut, and *gloverin* in the epidermis were highly expressed. Expression levels of several AMP genes were reduced after RNAi treatment. However, the susceptibility to treatment with RNAi specific to *Repat33* varied among four tissues. In the midgut, expression levels of most AMP genes except *apolipophorin III* were reduced after RNAi treatment. In the epidermis, expression levels of most genes except *attacin−2* and *gallerimycin* were reduced after RNAi treatment. In hemocytes, expression levels of *attacin−2*, *defensin*, two transferrin genes, and *cecropin* were reduced after RNAi treatment. In the fat body, expression levels of two *attacin* genes, *gallerimycin*, *lysozyme*, and *cecropin* were reduced after RNAi treatment. In all tissues, *cecropin* expression was highly susceptible to treatment with RNAi specific to *Repat33*. These results indicate that Repat33 can regulate tissue-specific expression of AMP genes.

### 3.4. Repat33 Expression Is Controlled by Eicosanoids

*Repat33* expression was inducible to bacterial challenge. However, treatment with dexamethasone known to inhibit PLA_2_ to block eicosanoid biosynthesis significantly (*p* < 0.05) suppressed the inducible expression of *Repat33* in response to bacterial challenge (Figure 5a). The addition of arachidonic acid (a catalytic product of PLA_2_) significantly rescued such inhibition of inducible *Repat33* expression caused by dexamethasone treatment. 

To see the influence of PLA_2_ on *Repat33* expression, RNAi specific to PLA_2_ was performed. The expression level of *Repat33* was then monitored. Results are shown in Figure 5b. As expected, bacterial challenge induced the expression of *Repat33*. However, RNAi specific for PLA_2_ expression significantly reduced *Repat33* expression even after bacterial challenge. However, the addition of arachidonic acid significantly (*p* < 0.05) rescued the inhibition of *Repat33* expression induced by treatment with RNAi specific for PLA_2_.

To clarify types of eicosanoids that mediated the induction of *Repat33* expression, two different eicosanoid biosynthesis inhibitors, ibuprofen (‘IBU’, prostaglandin (PG) biosynthesis inhibitor) and esculetin (‘ESC’, leukotriene biosynthesis inhibitor), were assessed. As shown in Figure 5c, both inhibitors significantly (*p* < 0.05) suppressed the inducible expression of *Repat33*, with the PG biosynthesis inhibitor being more potent. Addition of PGE_2_ significantly (*p* < 0.05) rescued *Repat33* expression. 

### 3.5. Repat33 Is a Downstream Component of PGE2 Signaling Pathway

The influence of PGE_2_ mediation on *Repat33* expression was analyzed in cellular and humoral immune responses (Figure 6). Hemocyte-spreading behavior induced by bacterial challenge was significantly inhibited by RNAi treatment against *Repat33* expression (Figure 6a). PGE_2_ also stimulated the hemocyte behavior. However, it failed to induce such cellular immune response under RNAi treatment specific for *Repat33*. 

Inducible expression of *cecropin* in four different tissues was commonly inhibited by RNAi of *Repat33* expression. This gene induction process was clarified by PGE_2_ mediation (Figure 6b). PGE_2_ without bacterial challenge significantly induced *cecropin* expression. However, RNAi specific to *Repat33* expression significantly suppressed the inducible gene expression of *cecropin*. Interestingly, such suppression was not rescued by the addition of PGE_2_. This indicates that PGE_2_ in response to the bacterial challenge can induce *Repat33* which mediates both cellular and humoral immune responses (Figure 6c).

## 4. Discussion

Repat family genes are known to be expressed in response to immune challenges in *S. exigua*. However, their physiological functions are little understood. This study investigated the role of Repat33, one of the Repat gene family members, in mediating cellular and humoral immune responses of *S. exigua*.

*Repat33* was predicted to be a secretory protein due to its signal peptide. It also possessed a multiprotein-bridging factor 2 (MBF2) domain with 34–108 residues. MBF2 can activate transcription via its interaction with a transcriptional factor IIA (TFIIA) [33]. In *Bombyx mori*, it is known to form a complex with MBF1 and a DNA-binding regulator FTZ-F1 [34]. This suggests that Repat33 might act as a co-activator to link specific and general TFs to induce specific immune genes. 

*Repat33* was highly expressed in the last larval instar of *S. exigua*. It was expressed in all tested tissues such as the gut, epidermis, hemocytes, and fat body of the last larvae instar. Its expression was up-regulated after bacterial infection. However, the inducible expression of *Repat33* after bacterial infection was suppressed by inhibition of eicosanoid biosynthesis. Injection of dexamethasone, a specific inhibitor of PLA_2_, significantly suppressed the expression of *Repat33*. However, the addition of arachidonic acid (AA), a catalytic product of PLA_2_, significantly rescued such suppression of its gene expression. This suggests that *Repat33* expression might be controlled by eicosanoids. This was further supported by RNAi treatment of PLA_2_ expression which suppressed the inducible expression of *Repat33* in response to bacterial challenge. The tested PLA_2_ was a secretory PLA_2_ (Se-PLA_2_) of *S. exigua*. Se-sPLA_2_ can mediate AA release to produce various eicosanoids [15]. Eicosanoids are known to mediate both cellular and humoral immune responses in insects [4]. Especially, this current study showed that PGE_2_ could mediate the inducible expression of *Repat33*. Thus, AA addition to larvae treated with RNAi specific to Se-sPLA_2_ expression significantly rescued the suppression of *Repat33* expression. PGE_2_ is one of the well-known eicosanoids in *S. exigua* and other insects [35]. Recently, its receptor has been identified in *Manduca sexta*. It has been shown that PGE_2_ can bind to its specific receptor and increase the cAMP level in target cells [25]. A highly similar PGE_2_ receptor (GenBank accession number: MN381016.1) was also found in *S. exigua*. cAMP can activate protein kinase A which then activates cAMP-responsive element-binding protein (CREB) [36]. CREB is then translocated into the nucleus to activate target genes. Based on this signal pathway, control of *Repat33* expression by eicosanoids might be explained by the cAMP signal pathway in target cells. Indeed, the current study showed that PGE_2_ could induce the expression of *Repat33*. The hypothesis of PGE_2_-CREB-Repat33 needs to be explored in a subsequent study.

RNAi of *Repat33* expression resulted in immunosuppression of *S. exigua*. Hemocyte-spreading behavior was significantly impaired by this RNAi treatment. Hemocyte-spreading behavior is required for most cellular immune responses of insects [5]. In *S. exigua*, the hemocyte behavior is mediated by eicosanoids, especially PGE_2_, by increasing actin cytoskeleton and stimulating aquaporin [35]. This suggests that Repat33 as a downstream factor of eicosanoids might play a crucial role in the cytoskeletal rearrangement of hemocytes. The impairment of hemocyte-spreading behavior induced by RNAi specific for *Repat33* expression led to a significant decrease in nodule formation after bacterial challenge. Such RNAi treatment also suppressed the up-regulation of AMP gene expression in response to bacterial challenge. Eicosanoids can mediate AMP gene expression in *Bombyx mori* and *Drosophila melanogaster* [37,38]. Indeed, a functional cross-talk between Toll/IMD pathways and PLA_2_ activation in *Tribolium castaneum* has been reported [39]. Under PLA_2_ inhibition, various eicosanoids can stimulate different immune-associated protein genes including AMPs in *S. exigua* [24]. These findings suggest that Repat33 can directly or indirectly activate AMP gene expression after an eicosanoid signal. 

In summary, this study analyzed the expression and physiological functions of Repat33 in the immunity of *S. exigua*. Repat33 can mediate both cellular and humoral immune responses under the eicosanoid signaling pathway. Suppression of *Repat33* expression can lead larvae to have an immunosuppressive state. This is the first study demonstrating that Repat family genes are involved in the regulation of immune responses.

## Figures and Tables

**Figure 1 insects-12-00449-f001:**
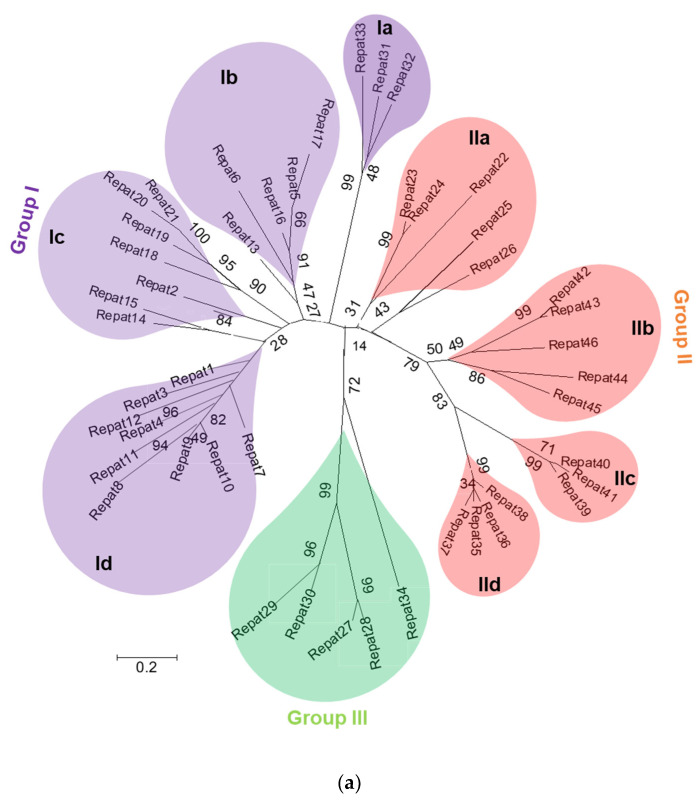
Classification of Repat family genes of *S. exigua*. (**a**) Phylogenetic analysis of 46 *Repat* genes. The analysis was performed using MEGA6. Bootstrapping values were obtained with 1000 repetitions to support branching and clustering. Amino acid sequences of selected genes were retrieved from GenBank with an accession number of PRJNA634227. (**b**) Expression of Repat genes induced by different pathogens. ‘Gram-’ and ‘Gram+’ represent *E. coli* and *E. mundtii* bacteria, respectively. Fungus and virus represent *M. rileyi* and AcMNPV, respectively. Immune challenge used a hemocoelic injection of bacteria (10^4^ cells/larva), fungus (1000 conidia/larva), or virus (10^4^ pfu/larva). After 8 h of incubation, total RNAs were extracted from the fat body and used for constructing cDNAs. A ribosomal gene, *RL32*, was used as a reference gene for RT-qPCR. Each treatment was replicated three times with independent tissue preparations. Different letters indicate significant differences among means at Type I error = 0.05 (LSD test).

**Figure 2 insects-12-00449-f002:**
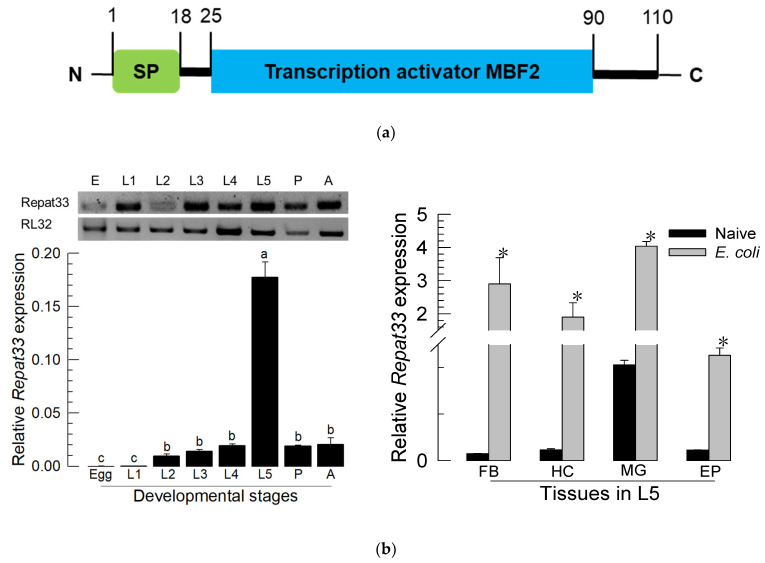
Expression profile of *Repat33* in *S. exigua*. (**a**) Domain analysis. ‘SP’ stands for signal peptide. (**b**) Expression levels of *Repat33* in different developmental stages including egg, first to fifth instar larvae (‘L1-L5′), pupa (‘P’), and adult (‘A’), and different tissues of L5 including the fat body (‘FB’), hemocyte (‘HC’), midgut (‘MG’), and epidermis (‘EP’). For the immune challenge, *E. coli* (10^4^ cells/larva) was injected into L5 larvae followed by incubation for 8 h at 25 °C. A ribosomal gene, *RL32*, was used as a reference gene. Each treatment was replicated three times with independent biological sample preparations. Different letters in developmental analysis indicate significant differences among means at Type I error = 0.05 (LSD test). Asterisks in tissue analysis indicate a significant difference between naïve and immune challenge.

**Figure 3 insects-12-00449-f003:**
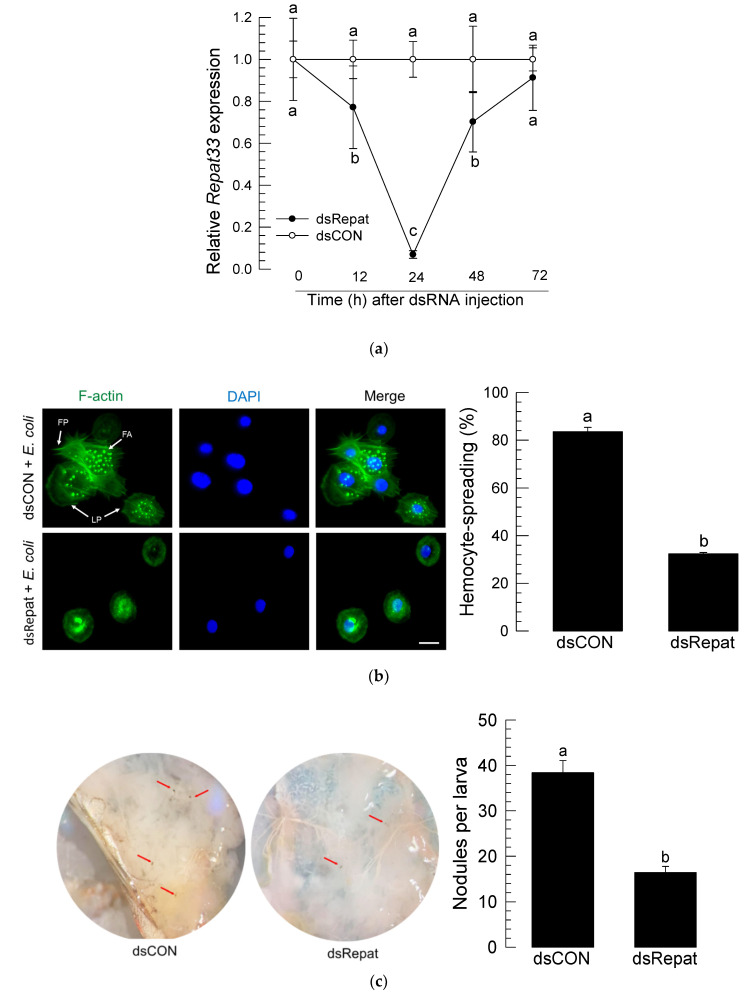
Effect of RNA interference (RNAi) of *Repat33* expression on cellular immune responses of L5 larvae of *S. exigua*. (**a**) Change of *Repat33* expression level in fat body after injection of dsRNA (‘dsRepat’, 1 µg/larva) specific to *Repat33*. A GFP gene was used to construct control dsRNA (‘dsCON’). Each treatment was replicated three times with independent tissue preparations. (**b**) Influence of RNAi on hemocyte-spreading behavior in response to bacterial challenge. *E. coli* (10^4^ cells/larva) was injected to L5 larvae at 24 h after injection of dsRepat. At 8 h after injection, hemolymph was collected and used for the hemocyte-spreading assay. For cytoskeleton analysis, hemocytes were observed under a fluorescence microscope at 400× magnification. Filopodia (‘FP’), focal adhesion (‘FA’), and lamellipodia (‘LP’) were indicated by white arrows. F-actin filaments were specifically recognized by FITC-tagged phalloidin (green). The nucleus was stained with DAPI (blue). Each treatment was independently replicated three times. Scale bar represents 10 µm. (**c**) Influence of RNAi on nodule formation in response to bacterial challenge (1.63 × 10^4^ cells/larva). Nodules were counted at 8 h post-injection. Each treatment was independently replicated three times. Different letters indicate significant differences among means at Type I error = 0.05 (LSD test).

**Figure 4 insects-12-00449-f004:**
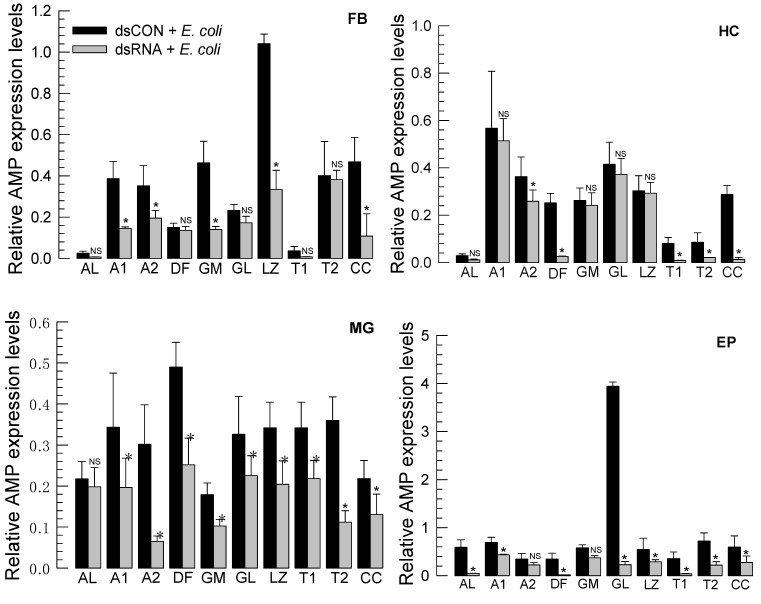
Effect of RNA interference (RNAi) of *Repat33* expression on humoral immune responses of L5 larvae of *S. exigua*. Humoral immune responses were assessed by quantifying expression levels of 10 antimicrobial peptides (AMP) genes: *apolipophorin III* (‘AL’), *attacin 1* (‘A1′), *attacin 2* (‘A2′), *defensin* (‘DF’), *gallerimycin* (‘GM’), *gloverin* (‘GL’), *lysozyme* (‘LZ’), *transferrin 1* (‘T1′), *transferrin 2* (‘T2′), and *cecropin* (‘CC’). L5 larvae were injected with dsRNA (‘dsRepat’, 1 µg/larva) specific for *Repat33*. At 24 h after injection, larvae were injected with *E. coli* (10^4^ cells/larva) and incubated at room temperature for 24 h. Total RNAs were extracted from different tissues [fat body (‘FB’), midgut (‘MG’), hemocyte (‘HC’), and epidermis (‘EP’)] and used for constructing cDNAs. A GFP gene was used to construct control dsRNA (‘dsCON’). Each treatment was replicated three times with independent tissue preparations. A ribosomal protein gene, *RL32*, was used as an internal control for RT-qPCR. Each treatment was replicated three times. Asterisks indicate significant differences exist (*p* < 0.05).

**Figure 5 insects-12-00449-f005:**
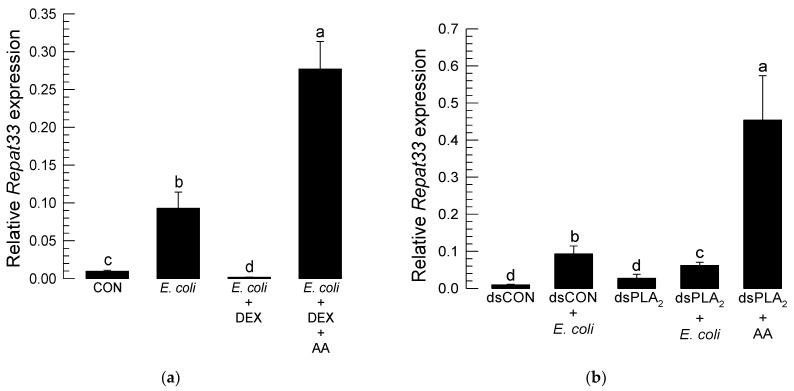
Influence of eicosanoids on *Repat33* expression in L5 larvae of *S. exigua*. For the immune challenge, *E. coli* (10^4^ cells/larva) was injected into L5 larvae followed by incubation at 25 °C for 8 h. (**a**) Inhibitory effect of dexamethasone (‘DEX’, a specific PLA_2_ inhibitor, 1 µg/larva) on *Repat33* expression. The addition of arachidonic acid (AA, a catalytic product of PLA_2_, 1 µg/larva) rescued *Repat33* expression suppressed by DEX. (**b**) Inhibitory effect of RNA interference (RNAi) of a PLA_2_ gene (= Se-sPLA_2_) of *S. exigua* on *Repat33* expression. RNAi was performed by injecting 1 µg of dsRNA (‘dsPLA2′) specific to Se-sPLA_2_ to each larva. A GFP gene was used to construct a control dsRNA (‘dsCON’). At 24 h after dsRNA injection, larvae were subjected to immune challenge. At 8 h after injection, total RNA was isolated from the fat body for RT-qPCR analysis of *Repat33* expression. (**c**) Effect of ibuprofen (‘IBU’, a COX inhibitor, 1 µg/larva) or esculetin (‘ESC’, a LOX inhibitor, 1 µg/larva) on *Repat33* expression. The addition of PGE_2_ (a catalytic product of COX, 1 µg/larva) rescued *Repat33* expression suppressed by IBU or ESC. A ribosomal protein gene, *RL32*, was used as a reference gene for RT-qPCR. Each treatment was replicated three times with independent biological sample preparations. Different letters indicate significant differences among means at Type I error = 0.05 (LSD test).

**Figure 6 insects-12-00449-f006:**
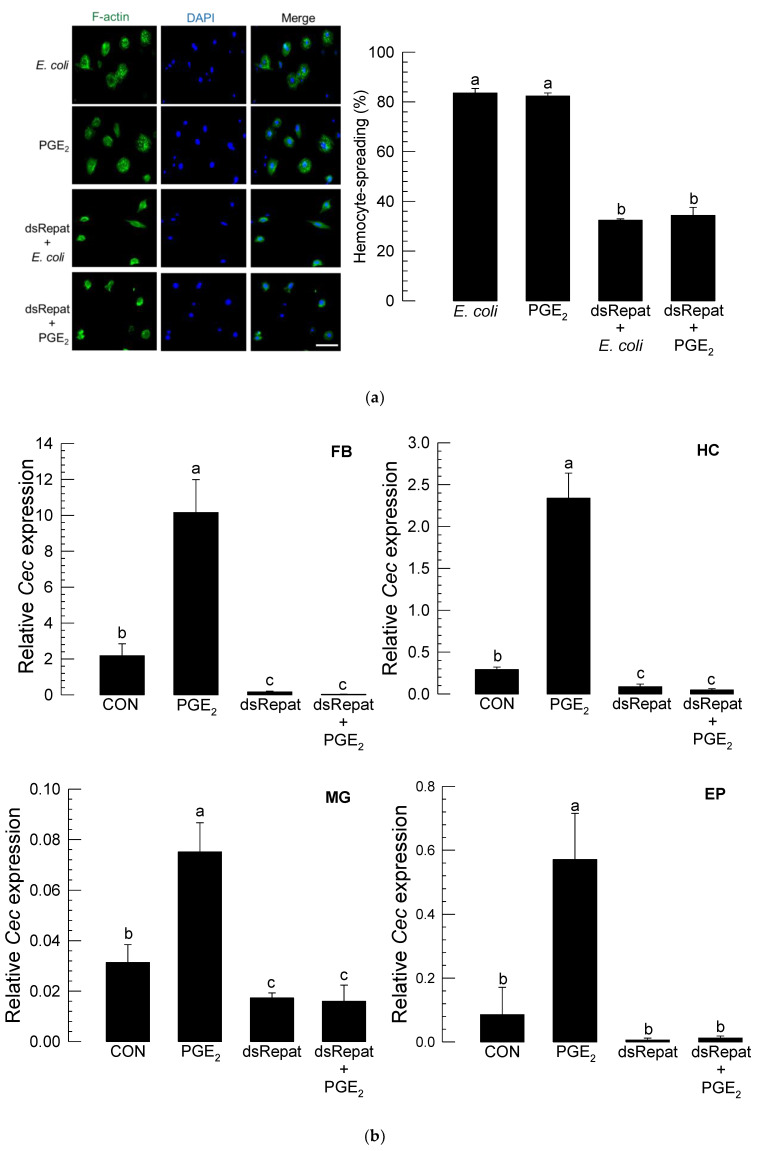
Repat33 as a downstream component of PGE_2_-mediating immune signal in *S. exigua*. (**a**) PGE_2_-Repat33 signaling in a cellular immune response assessed by hemocyte-spreading behavior. For the immune challenge, *E. coli* (10^4^ cells/larva) was injected into L5 larvae. After 8 h, hemolymph was collected and used for the hemocyte-spreading assay. PGE_2_ was injected at a dose of 1 µg/larva. RNA interference (RNAi) of *Repat33* expression was performed by injecting dsRNA (‘dsRepat’, 1 µg/larva) specific for *Repat33*. At 24 h after injection, larvae were treated with bacteria or PGE_2_. After 8 h, hemolymph was collected and used for the hemocyte-spreading assay. For cytoskeleton analysis, F-actin filaments were specifically recognized by FITC-tagged phalloidin (green). The nucleus was stained with DAPI (blue). Each treatment was independently replicated three times. Scale bar represents 10 µm. (**b**) PGE_2_-Repat33 signaling in humoral immune response assessed by measuring *cecropin* (‘Cec’) expression levels at 8 h after bacterial challenge. A ribosomal protein gene, *RL32*, was used as a reference gene for RT-qPCR. Each treatment was replicated three times with independent biological sample preparations. Different letters indicate significant differences among means at Type I error = 0.05 (LSD test). (**c**) Immune signaling pathway involving PGE_2_ and Repat33.

## Data Availability

Not applicable.

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
