# Peer review of "Repat33 Acts as a Downstream Component of Eicosanoid Signaling Pathway Mediating Immune Responses of Spodoptera exigua, a Lepidopteran Insect"

_insects, 2021, doi:10.3390/insects12050449_

Round 1
Reviewer 1 Report
The manuscript analyzed physiological functions of Repat33 in cellular and humoral immune responses of Spodoptera exigua. Repat33 is expressed in all developmental stages and induced in immune-associated tissues, larval hemocytes and fat bodies. Repat33 RNAi inhibited the hemocyte-spreading effect that is nodule formation of hemocytes against bacteria. In addition, the RNAi treatment down-regulated expression levels of some antimicrobial genes. Repat33 expression is regulated under eicosanoids using inhibition of eicosanoid biosynthesis by RNAi against a phospholipase A2 gene. Because adding arachidonic acid to RNAi treated insects made it recover from suppression of Repat33 expression. Authors suggested that Repat33 is a downstream component of eicosanoids for the immune responses in the moth. A lot of experimental evidence supports their hypothesis and conclusion in the paper that is well-written and would be recommended to publish. Authors proposed Repat33 mediate both pathways, cellular immunity and humoral immunity, I feel that there is a weak for the pathway of humoral immunity due to no direct evidence in the paper. Before acceptance, however, these issues pointed through the paper need to be revised or confirmed.
P2, L65 – reference citation.
P3, L107 – 37 °C
P3 Materials and Methods – need a brief method whether they conducted cloning of all the target genes or blast search all the sequences somewhere, then RT-PCT, qRT-PCR, and RNAi construction.
P3, L134: need to reword
P5, L215-216: Confirm whether it meant Repat33 gene or its product. Looks like 'Repat33 transcript is predicted to encode a small (110 amino acid residues) secretory protein …
P7, Fig1 legend: RJNA634227 couldn't find in GenBank; need to check the number. Should they have multiple numbers for all the Repat genes?
Authors picked a representative Repat gene from each group in Fig 1 that needed a rationale why they selected these genes somewhere M&M or Results.
P8, L267-269: needs to reword
Fig 4: Y-axis is confusing because target genes vary, not Repat33, compared to each APM gene.
Author Response
Comment #1-1: P2, L65 – reference citation.
Response: Corrected as [26]
Comment #1-2: P3, L107 – 37 °C
Response: Corrected
Comment #1-3: P3 Materials and Methods – need a brief method whether they conducted cloning of all the target genes or blast search all the sequences somewhere, then RT-PCT, qRT-PCR, and RNAi construction.
Response: As see below, a new subsection is added before the gene analysis.
“2.6. Bioinformatics analysis
- exigua Repat gene sequences were obtained from the transcriptomic databases (PRJNA634227). Prediction of protein domain structure was performed using Pfam (http://pfam.xfam.org) and Prosite (https://prosite.expasy.org/).”
Comment #1-4: P3, L134: need to reword
Response: Corrected as follows: “PCR product was assessed by 1% agarose gel electrophoresis.”
Comment #1-5: P5, L215-216: Confirm whether it meant Repat33 gene or its product. Looks like 'Repat33 transcript is predicted to encode a small (110 amino acid residues) secretory protein …
Response: Corrected as follows: “Based on the predicted amino acid sequence of Repat33, its product is a small…..”
Comment #1-6: P7, Fig1 legend: RJNA634227 couldn't find in GenBank; need to check the number. Should they have multiple numbers for all the Repat genes?
Response: Yes, the transcriptome contains 46 gene members. The accession number is PRJNA634227, which is double-checked.
Comment #1-7: Authors picked a representative Repat gene from each group in Fig 1 that needed a rationale why they selected these genes somewhere M&M or Results.
Response: Corrected as follows: “However, Repat33 was only highly inducible after infection of Gram-negative bacteria and used for subsequent functional assay in response to the bacterial immune challenge.”
Comment #1-8: P8, L267-269: needs to reword
Response: Corrected as follows: “The number of nodules formed in response to bacterial challenge was significantly (p < 0.05) decreased after treatment with Repat33 dsRNA compared to that after treatment with dsRNA control (Figure 3C).”
Comment #1-9: Fig 4: Y-axis is confusing because target genes vary, not Repat33, compared to each APM gene.
Response: Y-axis title is changed into “Relative AMP expression levels”
Reviewer 2 Report
Comments for Insect-1205451
In the manuscript, “Repat33 acts as a downstream component of eicosanoid signaling pathway mediating immune responses of Spodoptera exigua, a lepidopteran insect”, Hrithik and co-workers proposed that Repat33 plays an important role to link cellular immune response and humoral response against bacterial infection.
They found that Repat33 was highly expressed under Gram-negative bacterial infections. Further, the Repat33 was expressed in the developmental stage and different tissues under bacterial infection. Also, they predicted Repat33 may be a secreted protein based on the signal peptide of the N-terminal. dsRNA injection decreased the expression of mRNA of Repat33. Companied with inhibiting of cellular immune response by detection of hemocytes-spreading and nodule formation, and significantly decrease the mRNA level of antimicrobial peptide genes. Furthermore, PGE2 increased the expression of Cecropin but did not recover after dsRNA silence Repat33 under bacterial challenge, suggesting that Repat33 was a downstream component of the eicosanoid signaling pathway.
In general, the paper is interesting, but the writing of this manuscript is not clear, it is hard to understanding. And also, the logic of the manuscript is confusing. Thus, I believe this manuscript is not ready to publish. Some suggestions are listed below for the authors’ consideration.
- From Figure 1, there are at least two genes, Repth33 and Repath16, inducible after infection of Gram-negative bacteria. Not in P210, “However, Repat33 was only inducible after infection of Gram-negative bacteria”. Please clarify it.
- Repat33 was predicted to be a small secretory protein, but from Figure 2, there is no experiment to show Repat33 was a secretory protein. Need to add.
- Figure 4 is confusing, the expression of 10 antimicrobial peptides needs to redraw.
- The authors should read this manuscript carefully to correct some wrong descriptions, logic flow, and also English improvement required.
Author Response
Comment #2-1: From Figure 1, there are at least two genes, Repth33 and Repath16, inducible after infection of Gram-negative bacteria. Not in P210, “However, Repat33 was only inducible after infection of Gram-negative bacteria”. Please clarify it.
Response: Corrected as follows: “However, Repat33 was only highly inducible after infection of Gram-negative bacteria and used for subsequent functional assay in response to the bacterial immune challenge.”
Comment #2-2: Repat33 was predicted to be a small secretory protein, but from Figure 2, there is no experiment to show Repat33 was a secretory protein. Need to add.
Response: It is a prediction from a bioinformatics analysis. The sentence is revised as follows: “Based on the predicted amino acid sequence of Repat33, its product is a small (110 amino acid residues) secretory protein with signal peptide in its amino terminal region. Based on the predicted amino acid sequence of Repat33, its product is a small (110 amino acid residues) secretory protein with signal peptide in its amino terminal region.“
Comment #2-3: Figure 4 is confusing, the expression of 10 antimicrobial peptides needs to redraw.
Response: The figure is revised by changing Y-axis title as “Relative AMP expression levels”.
Comment #2-4: The authors should read this manuscript carefully to correct some wrong descriptions, logic flow, and also English improvement required.
Response: After all changes, the text is entirely read. The authors believe this revised version is improved to be more easily understandable.
Reviewer 3 Report
It is shown that the response to pathogen-33 (repat33) gene is involved in the regulation of the cellular and humoral immune response, in a tissue-specific manner in Spodoptera exigua. The Authors carry out a detailed analysis on the expression, inducibility, tissue-specific expression of Repat33 and define it as a downstream factor of eicosanoids. The involvement of the Repat proteins in the immune and in stress responses has been implicated; Repat33, as a secretory protein was clustered with stress response factors. As responses of the organism to immune stimulation and stress can affect each other and the invasion by microorganisms and parasites can also be considered to be as a stress factor. The studies clearly show that Repat is involved in a differential regulation of antimicrobial peptide production, aggregation and spreading of the blood cells, thus, providing evidence for being a component of a regulatory network in innate immunity.
Minor comments:
line 108: change 37 C to 37°C
line 168: use „permeabilized”, instead of „perforated”
line 169-170: „fluoreccein isothiocyanate (FITC) labelled(/tagged) phalloidin”, instead of „phalloidin-tagged fluorescein isothiocyanate”
line 170: manufacturer and used dilution of the conjugate is missing
line 172: „Hemocyte-spreading was observed” Pictures look nice, but how was it evaluated/scored/expressed?
line 216: change „transcriptional factor” to „transcription factor”
line 220: „all tested tissues showed…” which tissues were tested and which tissues were not inducible? Were all tissues induced?
line 403: „…as mentioned in Introduction” – omit
line 455-456: “This is the first study demonstrating that Repat family genes can mediate immune responses.” They are rather involved in regulation of…
Author Response
Comment #3-1: line 108: change 37 C to 37°C
Response: Corrected
Comment #3-2: line 168: use „permeabilized”, instead of „perforated”
Response: Corrected
Comment #3-3: line 169-170: „fluoreccein isothiocyanate (FITC) labelled(/tagged) phalloidin”, instead of „phalloidin-tagged fluorescein isothiocyanate” line 170: manufacturer and used dilution of the conjugate is missing
Response: Corrected as follows: “fluorescein isothiocyanate (FITC)-tagged phalloidin (Alexa Fluor 488 phalloidin, 1 µg/mL, Thermo Scientific, Rockford, IL, USA) in PBS at RT for 1 h.”
Comment #3-4: line 172: „Hemocyte-spreading was observed” Pictures look nice, but how was it evaluated/scored/expressed?
Response: The additional method details are added as follows: “Hemocyte-spreading was determined by the extension of F-actin out of the original cell boundary. Hemocyte-spreading behavior was quantified by randomly assessing 100 cells. Each treatment was replicated three times.”
Comment #3-5: line 216: change „transcriptional factor” to „transcription factor”
Response: Corrected
Comment #3-6: line 220: „all tested tissues showed…” which tissues were tested and which tissues were not inducible? Were all tissues induced?
Response: Yes, all tissues are induced.
Comment #3-7: line 403: „…as mentioned in Introduction” – omit
Response: Deleted
Comment #3-8: line 455-456: “This is the first study demonstrating that Repat family genes can mediate immune responses.” They are rather involved in regulation of…
Response: Corrected as follows: “Hemocyte-spreading was determined by the extension of F-actin out of the original cell boundary. Hemocyte-spreading behavior was quantified by randomly assessing 100 cells. Each treatment was replicated three times.”
Round 2
Reviewer 2 Report
The authors have addressed all of my concerns.